# Accentuated Eccentric Loading in the Bench Press: Considerations for Eccentric and Concentric Loading

**DOI:** 10.3390/sports9050054

**Published:** 2021-04-27

**Authors:** Christopher B. Taber, Jared R. Morris, John P. Wagle, Justin J. Merrigan

**Affiliations:** 1Department of Physical Therapy and Human Movement Science, Sacred Heart University, Fairfield, CT 06825, USA; morrisj154@mail.sacredheart.edu; 2Kansas City Royals, Kansas City, MO 64129, USA; johnwagle9@gmail.com; 3Rockefeller Neuroscience Institute, West Virginia University, Morgantown, WV 26505, USA; justin.merrigan@hsc.wvu.edu

**Keywords:** hierarchical linear modeling, eccentric training, weight releasers, augmented eccentric loading, velocity-based training

## Abstract

This study examined the effects of accentuated eccentric loading (AEL) on bench press velocities across a spectrum of concentric and eccentric loads. Ten strength trained men (bench press one-repetition maximum (1-RM): 124.3 ± 19.4 kg; relative strength ratio: 1.5 ± 0.2 kg∙body mass^−1^) participated. Subjects completed bench press repetitions using concentric loads from 30% to 80% 1-RM in 10% increments in each experimental session. The AEL protocols were implemented using 100% (AEL100) and 110% 1-RM (AEL110) loads during the eccentric action, while the eccentric load remained the same as the concentric for traditional loading (TRAD). Multilevel models analyzed the effects of each AEL protocol on concentric velocities across concentric loads (*p* < 0.05). Faster concentric velocities were observed at 30% 1-RM and 80% 1-RM with AEL100 compared to TRAD (*p* ≤ 0.05) but this effect was reduced for individuals moving the barbell through a greater displacement. Additionally, AEL110 presented a greater change in velocity from 30% to 80% 1-RM than TRAD (*p* ≤ 0.05). The AEL100 protocol resulted in faster concentric velocities throughout concentric loads of 30–80% 1-RM, but AEL110 may have been too great to elicit consistent performance enhancements. Thus, the efficacy of AEL at various concentric loads is dependent on the eccentric loading and barbell displacement.

## 1. Introduction

Post-activation performance enhancement manifested through faster bench press velocities at various concentric loads is an often-sought training outcome for strength-power athletes that require rapid pressing actions within their sport (e.g., American football offensive lineman; track and field shot putter). One method of acutely enhancing concentric velocity is accentuated eccentric loading (AEL), which is implemented by overloading the eccentric action during exercises involving the complete stretch shortening cycle (i.e., eccentric to concentric action) [1]. During the bench press exercise, AEL has demonstrated efficacy in potentiating acute performances compared to traditional loading (TRAD) where the eccentric load is the same as the concentric load [2,3,4,5,6]. However, divergent outcomes have been demonstrated, which may be partly attributed to differences in phasic loading and other variations between AEL protocols [2,4,6,7].

Maximal strength performances (one-repetition maximum, 1-RM) have been improved via AEL with 105% 1-RM [7], but were decreased using AEL with 105–120% 1-RM [4]. Thus, it is possible that the magnitude of the eccentric load influences the performance responses to AEL. However, little research has been conducted on differing magnitudes of AEL during submaximal concentric performances in the bench press. In one example, AEL did elicit a favorable effect on power output in the bench press with a fixed concentric load (50% 1-RM), while the corresponding optimal eccentric overload was unique to the individual [4]. Further, the concentric loads used when implementing AEL have been shown to dictate the efficacy of AEL for increasing concentric velocity [3]. Others found no change in concentric velocity when using AEL with 105% and concentric loads of 80% 1-RM [6]. Yet, AEL has been found to alter mechanics during the bench press which likely contributed to the loss in concentric velocity noted in AEL under the following eccentric to concentric loading patterns; 100%:30% 1-RM and 100%:80% 1-RM [8] and 120%:50% and 120%:65% [2]. Thus, further research is warranted to investigate AEL using multiple eccentric loads across a spectrum of concentric loads to identify the interaction of phasic loading parameters.

The greater load during the eccentric action has been shown to increase eccentric muscle activity, such that it is equivalent to the subsequent concentric action [9], providing rationale for the use of AEL. However, the increased muscle activity and forces during the eccentric action may not be enough to potentiate concentric performances during the bench press [8], although this may have been a result of altered technique. It has been speculated that stronger individuals are more capable of handling the heavier and more rapid movements during AEL [10]. As such, individuals with greater levels of relative strength have been less negatively affected by AEL [2]. Thus, exploring the loading schemes of AEL is likely best suited in resistance trained individuals that may better handle the excessive eccentric loading during AEL. Therefore, the purpose of this investigation was to explore the acute effects of different combinations of concentric and eccentric loading strategies during AEL bench press.

## 2. Materials and Methods

### 2.1. Subjects

Ten resistance trained men participated in the study (age: 23 ± 3 years, height: 175.5 ± 6.4 cm, body mass: 82.3 ± 9.2 kg, bench press one-repetition maximum (1-RM): 124.3 ± 19.4 kg; relative strength ratio: 1.5 ± 0.2). Subjects were required to have a minimum bench press relative strength ratio (1-RM divided by body mass) of 1.25 and had performed bench press consistently in training for the past year.

### 2.2. Study Design

A randomized counterbalanced design was used to examine the effects of AEL on bench press velocity across a spectrum of eccentric (100% and 110% 1-RM) and concentric loads (30%, 40%, 50%, 60%, 70%, and 80% 1-RM). During the first session, subjects completed 1-RM testing for prescription of eccentric and concentric loading in experimental sessions and were familiarized with AEL protocols. Following this initial session, three experimental conditions were conducted using either traditional loading (TRAD), accentuated eccentric load of 100% (AEL100), or 110% (AEL100) concentric maximum. All sessions were separated by one week and subjects were instructed to refrain from any upper body training 72-h prior to each testing session to minimize fatigue. Prior to each session, subjects completed a standardized 5-min dynamic warm-up for the upper body (e.g., jumping jacks, arm circles, and pushups).

### 2.3. One-Repetition Maximum (1-RM) Testing and Familiarization Session

All subjects completed a standardized dynamic warm-up followed by an incremental bench press warm up to determine the 1-RM, which is explained in further detail elsewhere [6]. Following the 1RM test, all subjects were familiarized with the weight releasers and were measured for appropriate height of the devices (i.e., the weight releaser disengaged immediately prior to the barbell touching the chest). Subjects were allowed to practice repetitions using the weight releasers to learn the technique of un-racking, steadying, and lowering the bar under control. Subjects were allowed up to five practice repetitions with weight to learn the technical aspects of the devices. The weight releaser measurements and rack heights were recorded and used for all subsequent sessions.

### 2.4. Experimental Sessions

Three randomized bench press loading protocols were implemented across a spectrum of concentric loads. Following the general dynamic warm up, the concentric loading for each experimental session was performed in consecutive order from 30, 40, 50, 60, 70, and 80% of the concentric 1-RM. One repetition was performed at each load and were all separated by 5 min of rest to limit fatigue. All subjects were instructed to perform each repetition as explosively as possible, with the intent to execute the concentric phase as fast as possible using a self-selected pace during the eccentric phase. TRAD was implemented using the same load during concentric and eccentric portions of the bench press. The additional eccentric load during AEL protocols was implemented using weight releasers (Monster Grips, Columbus, OH, USA) to equal 100% (AEL100) or 110% of the concentric 1-RM (AEL110) during the eccentric portion of the bench press. The weight releasers disengaged from the barbell upon hitting the ground allowing the concentric portion of the bench press during AEL protocols to be completed using the same concentric loads as traditional loading. Self-selected grip width was used by each participant and they were instructed to use the same grip width in each session. Subjects were instructed to keep feet flat on the floor, glutes, upper back, and head against the bench at all times during the execution of the lifts. The mean concentric velocity of each repetition was measured using a linear position transducer sampling at 50 Hz (GymAware Version 5: Kinetic Performance Technologies, Canberra, Australia) affixed to the barbell, which has been demonstrated to be valid and reliable [11]. Vertical displacement of each repetition was calculated from the bottom position of each repetition to the end of the concentric phase.

### 2.5. Statistical Analyses

Statistical procedures were performed in R version 3.6.2 (R Foundation, Vienna, Austria, https://www.R-project.org, accessed on March 2020) with an alpha level of *p* < 0.05. Data were not normally distributed. Thus, to understand the influence of AEL on mean concentric velocity, non-parametric hierarchical (multilevel) linear modeling approaches were used via the *‘nlme’ package* [12]. The repeated measures were assessed from concentric loads (level 1) and were nested within subjects (level 2). Explanatory predictor variables included concentric load, AEL100, AEL110, and displacement during the bench press. One model was conducted with 30% 1-RM as the intercept, while another was conducted with 80% 1-RM as the intercept to analyze the effects at the lightest and heaviest loads.

## 3. Results

Results from the multilevel models are presented in Table 1, group mean velocities are displayed in Figure 1, and individual data are presented in Figure 2. During 30% and 80% 1-RM concentric loads, AEL100 resulted in faster concentric velocities. However, as the displacement of the bar travel increased between individuals, the increase in concentric velocity due to AEL100 was decreased. When collapsed across all protocols, individuals achieving a greater displacement of the barbell achieved faster mean velocities during the bench press. Accentuated eccentric loading with 110% 1-RM resulted in a more negative slope of velocity across concentric loads from 30–80% 1-RM. Yet, concentric velocity during AEL110 was not significantly different than TRAD during 30% (*p* = 0.077) or 80% 1-RM (*p* = 0.543) concentric loads.

## 4. Discussion

This study examined the acute effects of AEL on concentric bench press velocities. The major findings were that (1) AEL100 produced faster concentric velocities in the bench press compared to TRAD; (2) AEL110 altered the slope of the load-velocity relationship from 30–80% of 1RM; and (3) greater barbell displacement was associated with faster concentric velocities and influenced the efficacy of AEL.

The AEL100 had faster concentric velocities across loads from 30–80% compared to TRAD, according to significant differences at 30% and 80% 1-RM and similar slopes between the loads. Although AEL110 resulted in similar concentric velocities as TRAD at 80% 1-RM concentric loads, the change in velocity from 80% to 30% 1-RM was greater. Resultantly, AEL110 resulted in a similar magnitude of increases in concentric velocity at 30% 1-RM, which neared significance, but failed to reach it due to a wide variation in responses to AEL110 (Figure 1 and Figure 2). This may suggest that the interaction between eccentric loads (110% vs. 100%) and concentric loads (30–80%) is necessary to consider. In prior bench press literature, AEL using 120% 1-RM decreased that repetition’s concentric velocity during concentric loads of 50% and 65% 1-RM [2]. Prior research implementing AEL using 120% 1-RM in the bench press and the back squat also found differences in the effectiveness of AEL depending on the concentric load [2,3]. These same studies also found a decrease in concentric velocity during the AEL repetition with 120% 1-RM; thus, the heavier eccentric loading may reduce the explosive abilities in the subsequent concentric phase [2,3]. However, when attempting to increase maximal force production, heavier eccentric loading during AEL (120% compared to 105%) would logically be more suited in driving that response, but little evidence supports an optimal eccentric overload [4]. Thus, the benefit of AEL may be dependent on the combination of the eccentric and concentric loading, which is subsequently influenced by the intention of the exercise (maximal force versus maximal velocity) and the size of the involved muscle groups (squat versus bench). It is likely the discrepancy in these findings is also in part related to the subjects involved in these studies and the differential strategies subjects use when lowering the bar with supramaximal loading. Although technique, such as pacing strategy, was self-selected, the researchers emphasized consistent execution of movement for each subject during each condition.

Moving the barbell through a greater displacement resulted in faster concentric velocities, but those moving through greater displacements noted a lower effect of AEL on concentric velocity. This finding indicates that subjects with different anthropometric qualities or technique may have differing responses to AEL. Though beyond the scope of the current investigation to elucidate specific mediating factors, it is possible barbell displacement variations due to differences in an individual’s arm length or grip width may influence acute response to AEL independent of phasic load. The application of AEL may alter lifting technique in the bench press and this has the potential to be magnified depending on the total system load and magnitude of loading on the eccentric portion of the exercise [2,8]. It is possible that in our current population the load of AEL110 was beyond their capabilities to maintain technique and could provide a reason for which concentric velocity was not enhanced. It appears that individual responsiveness to AEL plays a factor in improved concentric variables [2]. For this reason, practitioners should carefully monitor an athlete’s performance and quantify which loads will provide the optimal eccentric and concentric loading to improve bench press performance.

This study had limitations to consider. First, we did not measure the segment length of the arms for the subjects, which may have provided greater evidence on the effects of barbell displacement on AEL’s effectiveness. Secondly, heavier loads during AEL may reduce volitional eccentric velocities [2,3,13] and resultantly blunt potential immediate concentric performance enhancement. However, due to unreliable eccentric velocity data from the linear position transducer, these data were not presented in the current study.

## 5. Conclusions

This study provides necessary implications of loading schemes during bench press training with AEL, of which further research should be conducted to expound upon. Indeed, the effects of AEL are dependent on the combination of eccentric and concentric loading magnitudes, as AEL with 100% 1-RM enhanced concentric velocity more than AEL with 110% 1-RM. Yet, AEL with 110% 1-RM showed more viability for enhancing concentric performances with lighter concentric loads of 30% 1-RM compared to 80% 1-RM. The individual variation in responses to AEL should not go unnoticed and may partially be explained by the variations in barbell displacement.

## Figures and Tables

**Figure 1 sports-09-00054-f001:**
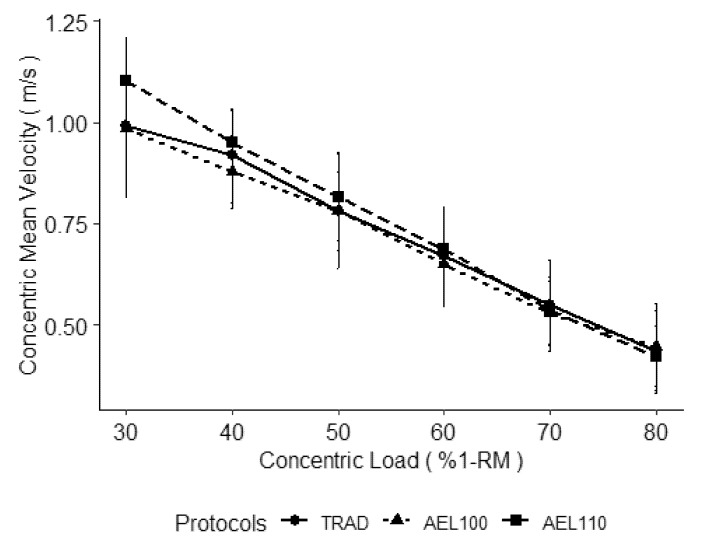
Mean concentric velocity of bench press repetitions with concentric loads from 30–80% of the bench press one-repetition maximum (%1-RM) using traditional loading (TRAD) and accentuated eccentric loading with 100% 1-RM (AEL100) and 110% 1-RM (AEL110). Data are presented as group mean with standard deviation as error bars.

**Figure 2 sports-09-00054-f002:**
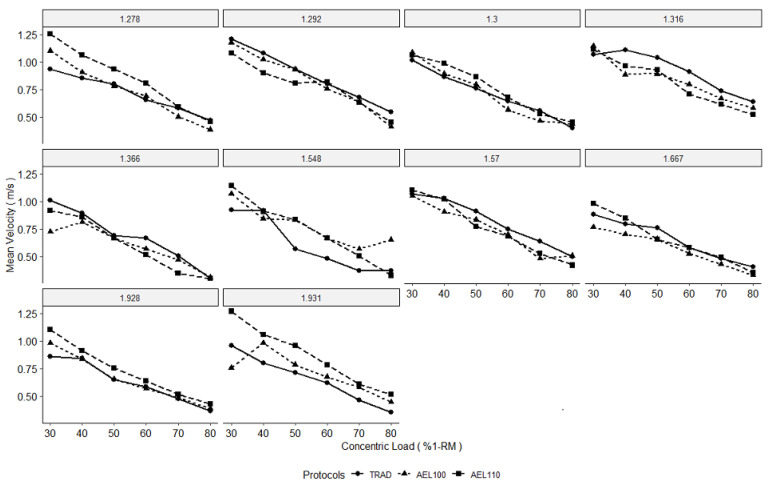
Mean concentric velocity of bench press repetitions with concentric loads from 30–80% of the bench press one-repetition maximum (%1-RM) using traditional loading (TRAD) and accentuated eccentric loading with 100% 1-RM (AEL100) and 110% 1-RM (AEL110). Data are displayed for each individual at each concentric load, with their coinciding relative maximal strength (1-RM/body mass) shown in the grey boxes above each subplot.

**Table 1 sports-09-00054-t001:** Multilevel models with fixed effects of concentric load and accentuated eccentric loading (AEL).

Factors	Loads from 30–80%	Loads from 80–30%
(Intercept)	0.267 (0.076) *	−0.249 (0.072) *
Displacement	0.048 (0.005) *	0.048 (0.005) *
Concentric Load	−0.103 (0.005) *	0.103 (0.005) *
Protocol AEL100	−0.002 (0.006)	0.002 (0.006)
Protocol AEL110	−0.025 (0.006) *	0.025 (0.006) *
Protocol AEL100	0.181 (0.074) *	0.169 (0.068) *
Protocol AEL100*Displacement	−0.014 (0.005) *	−0.014 (0.005) *
Protocol AEL110	0.183 (0.103)	0.059 (0.097)
Protocol AEL110*Displacement	−0.010 (0.006)	−0.010 (0.006)

Models were run with 30% and 80% concentric loading as the intercept. Results are displayed as the estimate (SE). *, indicates significant explanatory variable of the following meanings: Displacement, the effect of Displacement on velocity at 30% or 80%; Concentric Load, the slope of concentric loads from 30% or 80% one repetition maximum (1-RM) to 80% or 30% 1-RM, respectively; Indented Protocol AEL100, the effect of AEL100 on the slope of velocity from 30–80% and 80–30%; Indented Protocol AEL110, the effect of AEL110 on the slope of velocity from 30–80% and 80–30%; Protocol AEL100, the effect of AEL100 on velocity at 30% or 80%; Protocol AEL110, effect of AEL110 on velocity at 30% or 80%; * displacement on Protocols, effect of displacement on the effect of each protocol on 30% or 80% load.

## Data Availability

The data presented in this study are available on request from the corresponding author.

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
