# Peer review of "Accentuated Eccentric Loading in the Bench Press: Considerations for Eccentric and Concentric Loading"

_sports, 2021, doi:10.3390/sports9050054_

Round 1

Reviewer 1 Report

The study "Accentuated Eccentric Loading in the Bench Press: Considerations for Eccentric and Concentric Loading" aimed to compare different combinations of concentric and eccentric loadings on bench press exercise. 

The manuscript is well written and clearly presented, methods are correct.

I have just some minor comments:

  • Introduction, Line 67: CON and ECC have not been used anymore before and after this point in the manuscript, and the acronyms have not been explained. So, I would suggest changing with "concentric" and "eccentric" as it is throughout the manuscript.
  • Study Design: I found it not so easy to understand the study design, had to read it a couple of times. Maybe this part can be improved a bit to better explain that, after the first testing session, during the following three experimental sessions subjects performed the TRAD, AEL100 and AEL110, in a randomized order. Overall, the methods section can be made clearer and smoother in some parts.
  • Table 1. Please check formatting accordingly to the Journal guidelines.
  • Figures can be produced in a higher quality.

I believe this topic has a high practical relevance, and it is worthy of publication.

Given the above-reported comments,  I suggest a minor revision for this paper. 

Author Response

The manuscript is well written and clearly presented, methods are correct.

Thank you for your kind words.

I have just some minor comments:

Introduction, Line 67: CON and ECC have not been used anymore before and after this point in the manuscript, and the acronyms have not been explained. So, I would suggest changing with "concentric" and "eccentric" as it is throughout the manuscript.

Amended in the document, CON and ECC have been removed.

Study Design: I found it not so easy to understand the study design, had to read it a couple of times. Maybe this part can be improved a bit to better explain that, after the first testing session, during the following three experimental sessions subjects performed the TRAD, AEL100 and AEL110, in a randomized order. Overall, the methods section can be made clearer and smoother in some parts.

Amended in the document for clarity.

Table 1. Please check formatting accordingly to the Journal guidelines.

Formatting has been checked and updated

Figures can be produced in a higher quality.

Amended in the document

I believe this topic has a high practical relevance, and it is worthy of publication.

Given the above-reported comments, I suggest a minor revision for this paper. 

Thank you for your review and comments.

Reviewer 2 Report

General Comments: First I would like to thank the opportunity to review this interesting paper. The study investigated the effects of accentuated eccentric loading on bench press velocities in different concentric and eccentric loads. In general, the study is really interesting and properly written. The authors must be acknowledged for it. The introduction and methods sections are well presented and the results are clear. The discussion section is interesting and based on the resutls of this and previous research. There are just some minor aspects that I think should be addressed before I can endorse the publication of the study. Specific comments below.

Title

No comments

Abstract 

No comments

Introduction

The introduction is well written, interesting and adequately presents the research question.

Line 48. Please replace "the concentric loads used when implementing AEL has been" with "the concentric loads used when implementing AEL have been"

Methods

Line 104. Why was a self-selected pace allowed? Wouldn`t that potenially influence the subsequent concentric phase? How did the authors control for these differences (i.e., fast eccentric phases potentially eliciting the stretch shortening cycle to a greater extent)? Also, can the authors provide data from the velocity in the eccentric phase of the movement with the different loading conditions?

Line 124. How was the displacement during the bench press determined? Please clarify in the methods section.

Results

In Figure 2, please clarify what the numbers inside the grey squares represent. 

Discussion

Line 170. "which neared significance, but failed to reach significance". Consider replacing the second word "signifcance to avoid repetition: "which neared significance, but failed to reach it

Line 185-187. "It is likely the discrepancy in these findings is also in part related to the subjects involved in these studies and the differential strategies subjects use when lowering the bar with supramaximal loading."

I consider this to be an important aspect that may have potentially influenced the results. Can the authors confirm that the stategy used in the eccentric phase by each individual was the same in AEL110, AEL100 and TRAD? This should be further discussed in the manuscript.

Line 197-202. "It is possible that in our current population the load of AEL110 was beyond their capabilities to maintain technique and could provide a reason for which concentric velocity was not enhanced. It appears that individual responsiveness to AEL plays a factor in improved concentric variables [2]. For this reason, practitioners should carefully monitor an athlete’s performance and quantify which loads will provide the optimal eccentric and concentric loading to improve bench press performance.

Great point here. I really like how the authors interpreted the data and provide practical solutions to coaches. Congratulations.

Line 203-205. "First, we did not measure the segment length of the arms for the subjects which may have provided greater evidence on the effects of barbell displacement on AEL’s effectiveness". 

It is still not clear to me how displacement was determined. This is a crucial point that should be explained in a revised version of the manuscript.

Line 211-216. I recommend the authors to present this section of the text in a separate section: Conclusion.

I acknowledge that this section is optional as per journal styling but, for the reader, it would be beneficial. This is a personal opinion and the authors shoudl not feel obliged to change the manuscript based on this comment.

Author Response

General Comments: First I would like to thank the opportunity to review this interesting paper. The study investigated the effects of accentuated eccentric loading on bench press velocities in different concentric and eccentric loads. In general, the study is really interesting and properly written. The authors must be acknowledged for it. The introduction and methods sections are well presented and the results are clear. The discussion section is interesting and based on the resutls of this and previous research. There are just some minor aspects that I think should be addressed before I can endorse the publication of the study. Specific comments below.

Thank you for your kind words and reviewing our manuscript.

Title

No comments 

Abstract 

No comment

Introduction

The introduction is well written, interesting and adequately presents the research question.

Line 48. Please replace "the concentric loads used when implementing AEL has been" with "the concentric loads used when implementing AEL have been"

Amended in the document.

Methods

Line 104. Why was a self-selected pace allowed? Wouldn`t that potenially influence the subsequent concentric phase? How did the authors control for these differences (i.e., fast eccentric phases potentially eliciting the stretch shortening cycle to a greater extent)? Also, can the authors provide data from the velocity in the eccentric phase of the movement with the different loading conditions?

This is a great question. They were allowed to self-select pace due to the discrepancy in weight on the bar. It was very challenging on the first two reps to stabilize the weight for some of the subjects so we allowed them to lower at their chosen pace for safety. While conducting the study we attempted as best as we could for them to use the same technical execution while accounting for individual differences. Finally, we did have data from the eccentric phase. However, we did not trust the reliability of the data from the device and we did not include it in our analysis.

Line 124. How was the displacement during the bench press determined? Please clarify in the methods section.

Amended in the document, Lines 119-120.

Results

In Figure 2, please clarify what the numbers inside the grey squares represent.

Amended in the figure caption, these numbers were relative strength values.

Discussion

Line 170. "which neared significance, but failed to reach significance". Consider replacing the second word "signifcance to avoid repetition: "which neared significance, but failed to reach it

This change has been made.

Line 185-187. "It is likely the discrepancy in these findings is also in part related to the subjects involved in these studies and the differential strategies subjects use when lowering the bar with supramaximal loading."

I consider this to be an important aspect that may have potentially influenced the results. Can the authors confirm that the stategy used in the eccentric phase by each individual was the same in AEL110, AEL100 and TRAD? This should be further discussed in the manuscript.

The authors can confirm the subjects were instructed to maintain the similar technique on all conditions. The main difference between the traditional loading and the AEL is the stabilization at the top of each rep. While each subject may have used a slightly different strategy we emphasized using the same technical execution on each repetition. We tried to use stronger individuals to prevent between rep fluctuations.

 We have added a line after 194-195 to reflect this.

Line 197-202. "It is possible that in our current population the load of AEL110 was beyond their capabilities to maintain technique and could provide a reason for which concentric velocity was not enhanced. It appears that individual responsiveness to AEL plays a factor in improved concentric variables [2]. For this reason, practitioners should carefully monitor an athlete’s performance and quantify which loads will provide the optimal eccentric and concentric loading to improve bench press performance.

Great point here. I really like how the authors interpreted the data and provide practical solutions to coaches. Congratulations.

 Thank you for this comment.

Line 203-205. "First, we did not measure the segment length of the arms for the subjects which may have provided greater evidence on the effects of barbell displacement on AEL’s effectiveness". 

It is still not clear to me how displacement was determined. This is a crucial point that should be explained in a revised version of the manuscript.

 We have explained in the methods now how displacement was calculated from the gymaware unit. This should help to clarify this point.

Line 211-216. I recommend the authors to present this section of the text in a separate section: Conclusion.

I acknowledge that this section is optional as per journal styling but, for the reader, it would be beneficial. This is a personal opinion and the authors should not feel obliged to change the manuscript based on this comment.

We have made the suggested change in the manuscript.

Reviewer 3 Report

This study aims to test the effect of accentuated eccentric loading on concentric performance during bench press. The article is well constructed, introduction and conclusion sections are clear and succinctly present the current knowledge on post-activation performance enhancement. However the method and results sections lacks of clarity and further details should be presented to better consider the findings of the present study (see my specific comments below). Furthermore it could be interesting to present values of measured variables (e.g., mean velocity) in order to facilitate reader’s comprehension and appropriation of the results.

Methods

L 102-104: please precise if subjects have to stop the barbell on the chest between the eccentric and the concentric phase to avoid a “plyometric-like” movement, or whether the subjects were free to sequence this phase as they want? Furthermore, it is surprising that no recommendation was provided for the eccentric phase of the movement, which is the focus of this study. Different time under tension during this phase may be expected across the protocols, that would likely results to different acute neural or structural adaptations (e.g. Chapman et al., 2006; 27(8); IJSM, Cintineo et al., 2018; 32(12); JSCR) and ultimately influence the subsequent concentric phase. Please justify this choice.

L 111-112: the absence of specific hand position would have obviously conducted to different shoulder and elbow joints configurations when barbell touch the chest that would ultimately affect muscle length and force development capacities. How did the authors consider this parameter during analysis? Furthermore, what about legs and feet positions and use? Did the participants were free to use their legs and lift their back while pushing?

Results

This section lacks of clarity. Please justify why the authors compared the relation from 30 to 80% and the inverse relation –i.e. from 80% to 30%)? Furthermore, it could be suitable to also report differences on figure 1 to ease understanding of the present findings.

L127-130; could you report mean velocities in the text with statistical power? Please report significant differences in mean barbell velocity on the figure 1 for a better clarity of the results.

L 134: did TL refers to traditional? If so, please rename in accordance with the methods section (i.e. TRAD).

Fig 2: should reconsider the use of this figure. It does not provide a clear description of the different profiles of the athletes or further reliable details to understand findings.

Discussion:

L 168: your findings showed similar slopes, therefore, would it be rather only greater velocity reported for AEL110 compared to TRAD rather than greater change in velocity? Which could means that velocity would vary from a greater extent between loads.

L 179: it seems that the GymAware allows calculation of the rate of force development. Did this variable could be calculated from your findings, that would allow you discuss about change in explosivity?

L 190-191: in light of this hypothesis, authors should provide further details about shoulder and elbow joints configuration, legs positions and whether participants lift their back during movement.

Author Response

This study aims to test the effect of accentuated eccentric loading on concentric performance during bench press. The article is well constructed, introduction and conclusion sections are clear and succinctly present the current knowledge on post-activation performance enhancement. However the method and results sections lacks of clarity and further details should be presented to better consider the findings of the present study (see my specific comments below). Furthermore it could be interesting to present values of measured variables (e.g., mean velocity) in order to facilitate reader’s comprehension and appropriation of the results.

Thank you for your review and your comments.

Methods

L 102-104: please precise if subjects have to stop the barbell on the chest between the eccentric and the concentric phase to avoid a “plyometric-like” movement, or whether the subjects were free to sequence this phase as they want? Furthermore, it is surprising that no recommendation was provided for the eccentric phase of the movement, which is the focus of this study. Different time under tension during this phase may be expected across the protocols, that would likely results to different acute neural or structural adaptations (e.g. Chapman et al., 2006; 27(8); IJSM, Cintineo et al., 2018; 32(12); JSCR) and ultimately influence the subsequent concentric phase. Please justify this choice.

We allowed the subjects to self-select the eccentric pacing due to the difficult nature of AEL. The first two conditions provided quite a challenge for the subjects and we allowed them to control the weight with their own self-selected technical execution. We did however instruct them to use the same execution of each rep. So once they stabilized the weight releasers whatever speed they chose to use on the eccentric was expected to be used each time and we constantly reminded through the trails. Lines 188-190 were added based on recommendations from the reviewers. We also intended to analyze the velocity of the eccentric phase to support these hypotheses but we did not trust the reliability of these data.

L 111-112: the absence of specific hand position would have obviously conducted to different shoulder and elbow joints configurations when barbell touch the chest that would ultimately affect muscle length and force development capacities. How did the authors consider this parameter during analysis? Furthermore, what about legs and feet positions and use? Did the participants were free to use their legs and lift their back while pushing?

We instructed the subjects to use the same hand position each time. This was self-selected by the subject in the position that was most comfortable and allowed them to lift the most weight. The same bar was used for subjects so they knew where to line up their hands each time. The subjects were advanced lifters who had stable technique so they had familiarity with what hand position to use. We have added a line about back and feet position. They were expected to maintain contact points at all times a line was added at 114-116 to reflect this. It would be considered a no rep if their hips came of the bench when lifting. Because the highest concentric value encountered in this study was 80% no subject had to lift their hips on the concentric portion of the lift.

Results

This section lacks of clarity. Please justify why the authors compared the relation from 30 to 80% and the inverse relation –i.e. from 80% to 30%)? Furthermore, it could be suitable to also report differences on figure 1 to ease understanding of the present findings.

These models investigated the slope from 30- 80% concentric loads, thus we cannot add post hoc comparisons of each differences at loads in figure 1. We feel this analysis is most appropriate due to the distribution of the data. However, due to the results of the slope differences we ran a second model to determine the difference at the intercept at 80% 1-RM to investigate if the differences in slopes resulted in a difference in results at the lightest and heaviest concentric loads. However, despite this difference in slopes of AEL110 there was no effect of AEL110 on either 30 or 80% concentric load (likely due to large individual variations in response at 30%) We have added a sentence from lines 128 to 130 on the reason for looking at 30% nd 80% 1-RM for the intercept in these models.

L127-130; could you report mean velocities in the text with statistical power? Please report significant differences in mean barbell velocity on the figure 1 for a better clarity of the results.

Unfortunately, we cannot add to the figure to show significant differences at each load since these statistical analyses did not investigate the overall mean difference and post hoc comparisons noted in traditional anova protocols. However, for reasons mentioned above we feel this analysis is most appropriate considering individual differences in responses and distribution of the data.

L 134: did TL refers to traditional? If so, please rename in accordance with the methods section (i.e. TRAD).

Amended in the document.

Fig 2: should reconsider the use of this figure. It does not provide a clear description of the different profiles of the athletes or further reliable details to understand findings.

Thank you for this idea, after some consideration we have decided to leave this figure as we feel showing the individual responses to these protocols may be valuable for future investigators to consider.

Discussion:

L 168: your findings showed similar slopes, therefore, would it be rather only greater velocity reported for AEL110 compared to TRAD rather than greater change in velocity? Which could means that velocity would vary from a greater extent between loads.

Ah yes, the slope was similar for AEL100 but the slope was not similar for AEL110. Thus, AEL110 resulted in a greater change in velocity from 30-80% but likely did not result in significant differences at 30% due to the large individual variations in responses.

L 179: it seems that the GymAware allows calculation of the rate of force development. Did this variable could be calculated from your findings, that would allow you discuss about change in explosivity?

The eccentric variables and rate of force development were deemed unreliable and were not included on our analysis.

L 190-191: in light of this hypothesis, authors should provide further details about shoulder and elbow joints configuration, legs positions and whether participants lift their back during movement.

We have expanded the methods to account for hand position and technical execution in order to clarify these points.

Thank you for your thorough review of our paper and your comments which have improved the manuscript.